# Shape–Preserved CoFeNi–MOF/NF Exhibiting Superior Performance for Overall Water Splitting across Alkaline and Neutral Conditions

**DOI:** 10.3390/ma17102195

**Published:** 2024-05-07

**Authors:** Yu Liu, Panpan Li, Zegao Wang, Liangjuan Gao

**Affiliations:** College of Materials Science and Engineering, Sichuan University, Chengdu 610065, China; 2021223010040@stu.scu.edu.cn (Y.L.); panpanli@scu.edu.cn (P.L.); zegao@scu.edu.cn (Z.W.)

**Keywords:** electrocatalysis, hydrogen evolution reaction (HER), oxygen evolution reaction (OER), overall water splitting, metal–organic framework (MOF), alkaline and neutral conditions

## Abstract

This study reported a multi–functional Co_0.45_Fe_0.45_Ni_0.9_–MOF/NF catalyst for oxygen evolution reaction (OER), hydrogen evolution reaction (HER), and overall water splitting, which was synthesized via a novel shape–preserving two–step hydrothermal method. The resulting bowknot flake structure on NF enhanced the exposure of active sites, fostering a superior electrocatalytic surface, and the synergistic effect between Co, Fe, and Ni enhanced the catalytic activity of the active site. In an alkaline environment, the catalyst exhibited impressive overpotentials of 244 mV and 287 mV at current densities of 50 mA cm^−2^ and 100 mA cm^−2^, respectively. Transitioning to a neutral environment, an overpotential of 505 mV at a current density of 10 mA cm^−2^ was achieved with the same catalyst, showing a superior property compared to similar catalysts. Furthermore, it was demonstrated that Co_0.45_Fe_0.45_Ni_0.9_–MOF/NF shows versatility as a bifunctional catalyst, excelling in both OER and HER, as well as overall water splitting. The innovative shape–preserving synthesis method presented in this study offers a facile method to develop an efficient electrocatalyst for OER under both alkaline and neutral conditions, which makes it a promising catalyst for hydrogen production by water splitting.

## 1. Introduction

In recent years, environmental issues, especially the greenhouse effect due to fossil fuel combustion, have garnered substantial attention since environmental conservation has significantly intensified [1]. Therefore, the paradigm is shifting from reliance on non–renewable sources to a broader acceptance of clean, renewable energies, such as solar energy, wind energy, hydrogen energy, geothermal energy, and so on. Among these, hydrogen energy stands out as a sustainable and environmentally benign energy carrier [2]. Fuel cell is one of the most effective means to utilize hydrogen, which directly converts chemical energy into electricity at a remarkably high efficiency without generating any pollution in the whole process, ensuring zero environmental contamination [3]. However, the cost of hydrogen production is one of the reasons that impedes the practical application of fuel cells [4,5]. Therefore, seeking a cost–effective method to produce hydrogen at a large scale is imperative. Water electrolysis is considered as one of the most effective solutions to produce hydrogen since water is abundant on Earth in rivers and seas [6,7]. In addition, the electricity generated by solar cells/power and wind could be used as the source of electricity to drive the water electrolysis, which would mitigate the depletion of fossil fuels and reduce the emission of CO_2_ to the environment [8,9,10]. However, an overpotential is required during water electrolysis, which means that the potential required for water electrolysis is higher than the thermodynamic standard potential (1.23 V) due to interfacial resistance, concentration polarization, etc. [11,12]. Thus, the design and preparation of an efficient electrocatalyst to minimize the overpotential during the water electrolysis has attracted great attention.

Nowadays, water electrolysis under alkaline environments has been commercialized to produce hydrogen due to several advantages. Predominantly, the lower overpotential under alkaline conditions allows for operation at higher current densities, resulting in the reduction in energy consumption and lifespan extension of the electrolysis equipment, thereby ensuring efficient and stable hydrogen production [13]. Secondly, the restriction on catalyst materials under alkaline conditions is not as stringent as that under acidic conditions. Many non–precious metals such as transition metals and their oxides could be used as potential alkaline water electrolysis catalysts, reducing the application cost of catalysts [14]. Thirdly, the electrolytic tank of alkaline water splitting has a simple structure and low cost because it does not need expensive components such as proton exchange membranes [15]. However, the corrosion of catalysts under alkaline conditions has directed the attention of water electrolysis towards neutral conditions in recent years. Neutral solutions are appealing due to their straightforward preparation, reduced risks, and environmental compatibility [16]. Yet, challenges persist, including a sluggish mass transfer rate, lower current densities, elevated overpotentials, compromised stability, and durability, along with limited comprehension of the reaction mechanisms and kinetics under neutral conditions [17]. In essence, developing a more efficient, durable, and cost–effective electrocatalyst holds profound implications for advancing hydrogen production via water electrolysis, particularly under alkaline and neutral conditions [18].

Metal–organic frameworks (MOFs) represent an emerging class of materials, which possess a stable porous architecture by linking metal atoms with organic ligands via covalent bonds and other interactions [19]. Distinct from traditional materials, MOFs captivate attention due to their low density, extensive specific surface area, and wealth of metal active centers [20]. However, their suboptimal electrical conductivity and inherent catalytic activity have curtailed their potential application in catalytic regions. Therefore, researchers developed preparation strategies such as synergistic effect [21,22], morphology control [23], structure engineering [24], defect engineering [22,25], etc. to make MOF–based catalysts more adaptable to the catalysis of OER and HER. In addition, heterometallic catalysis has been studied extensively [26]. The synergistic effect between heterometallic cations also provides a new idea for the preparation of MOF catalysts [21,27]. For example, Cheng et al. prepared nanobox–like CoCu–MOF through successive cation and ligand exchange strategies, and the synergistic effect between different metal elements and the enhanced active site exposure make it obtain excellent OER performance and stability [28]. Tan et al. loaded ultrafine Ni particles into MOF by partial carbonization, and the rod–like morphology increased the electrochemical active area and the number of active sites of the catalyst. The addition of Ni provides a synergistic effect, enhancing the electrical conductivity of MOF, and enabling HER catalysis in alkaline media to drive a current density of 10 mA cm^−2^ with an overpotential of only 131 mV [29]. Ni–MOF has excellent catalytic activity and electrochemical durability as an electrocatalyst material. However, the overpotential of monomeric MOFs is not satisfactory for rapidly driving OER processes [30,31]. Sun et al. synthesized Fe–doped NiCo–MOF (Fe–NiCo–MOF/NF) by solvothermal method. The changed coordination surroundings changed the local electronic structures of Ni and Co, which could improve the catalytic performance. Because of the enhanced intrinsic activity by the synergistic effect of heterometallic cations, the Fe–NiCo–MOF/NF needs only an overpotential of 290 mV to reach 50 mA cm^−2^ for OER [32].

At present, the one–step hydrothermal method and annealing are commonly used to synthesize MOF–based catalysts. However, the morphology of MOFs obtained by these methods might not meet experimental expectations, resulting in the overuse of raw materials and delays in the development of new catalysts [33,34]. For example, Deng et al. synthesized Ni–MOF by the simple one–step hydrothermal method. However, the generated Ni–MOF exhibits a stratified but randomly oriented morphology, which might result in a low specific surface area and few active sites, thus damaging the catalytic properties of MOF [35]. At the same time, few existing MOF–based catalysts have been shown to exhibit excellent overall water splitting performance, especially for catalytic applications across neutral and alkaline conditions. Therefore, it is important to develop effective strategies to synthesize MOF–based electrocatalysts for making use of the advantages of MOF structures and improve the overall performance of such catalysts under both neutral and alkaline conditions. 

In this study, we synthesized MOF–based electrocatalysts, incorporating metal ions of Co, Fe, and Ni, via the shape–preserving hydrothermal method, which exhibited high catalytic activity. Figure 1 shows the illustration of the fabrication strategy and process from layered double hydroxide (LDH) to MOF. Remarkably, benefitting from the excellent catalytic activity brought by the synergistic effect of Co, Fe, and Ni cations and the abundant active sites provided by the unique bowknot flake structure, an overpotential of only 287 mV was required to achieve a current density of 100 mA cm^−2^ for OER under 1 M KOH condition, while only 164 mV for HER at a current density of 10 mA cm^−2^ when the CoFeNi–MOF/NF catalyst was used. Under 1 M PBS condition, the overpotentials required for OER and HER at a current density of 10 mA cm^−2^ were 505 mV and 241 mV, respectively. To simulate industrial conditions, a two–electrode electrolytic cell was assembled to evaluate the comprehensive electrolysis performance of CoFeNi–MOF/NF, which demonstrated that such a catalyst exhibited notable catalytic activity. Furthermore, the performance of the catalyst did not degrade significantly across the full process, showing high catalytic stability. This study has provided an effective strategy for designing and crafting innovative bifunctional catalysts that are suitable for diverse environments.

## 2. Experimental

### 2.1. Materials

Ethanol (C_2_H_5_OH, AR), hydrochloric acid (HCl, AR), nickel nitrate hexahydrate (Ni(NO_3_)_2_·6H_2_O, AR), iron nitrate nonahydrate (Fe(NO_3_)_2_·9H_2_O, AR), cobalt nitrate hexahydrate (Co(NO_3_)_2_·6H_2_O, AR), ammonium fluoride (NH_4_F, AR), urea (CO(NH_2_)_2_, AR), N, N–dimethylformamide (DMF, AR), di–potassium hydrogen phosphate trihydrate (K_2_HPO_4_·3H_2_O, AR), and potassium dihydrogen phosphate (KH_2_PO_4_, AR) were purchased from Chengdu Kelong Chemical Co., Ltd. (Chengdu, China). p–Phthalic acid (H_2_BDC, 99%) was purchased from Shanghai Aladdin Biochemical Technology Co., Ltd. (Shanghai, China). Potassium hydroxide–water standard titration solution (KOH 1.0000 mol/L) was purchased from Guangzhou Howei Pharma Tech Co., Ltd. (Guangzhou, China). Nickel foam (NF) was purchased from Shenzhen Green and Creative Environmental Science and Technology Co., Ltd. (Shenzhen, China), and was pretreated with HCl, deionized water, and ethanol several times to remove the surface impurities.

### 2.2. Preparation of CoFeNi–LDH/NF

Typically, 0.45 mmol Fe(NO_3_)_2_·9H_2_O, 0.9 mmol Ni(NO_3_)_2_·6H_2_O, 0.45 mmol Co(NO_3_)_2_·6H_2_O, 3 mmol NH_4_F, and 6 mmol urea were dissolved into 80 mL deionized water and stirred for 15 min. Then, the solution was transferred into a 100 mL Teflon–lined autoclave. A piece of pretreated NF (3 cm × 4 cm) was immersed into the above solution. The autoclave was then heated at 120 °C for 6 h. After cooling down to room temperature, the sample was washed several times with deionized water and ethanol and then dried at 70 °C for 3 h. The sample was named Co_0.45_Fe_0.45_Ni_0.9_–LDH/NF.

### 2.3. Preparation of CoFeNi–MOF/NF 

In this procedure, 0.45 mmol Fe(NO_3_)_2_·9H_2_O, 0.9 mmol Ni(NO_3_)_2_·6H_2_O, 0.45 mmol Co(NO_3_)_2_·6H_2_O, and 1.35 mmol H_2_BDC were dissolved into 70 mL DMF and stirred for 15 min. Then, 5 mL deionized water and 5 mL ethanol were added into the solution and stirred for 10 min. Then, the solution was transferred into a 100 mL Teflon–lined autoclave. The Co_0.45_Fe_0.45_Ni_0.9_–LDH/NF was immersed into the above solution. The autoclave was then heated at 120 °C for 12 h. After cooling down to room temperature, the sample was washed several times with deionized water and ethanol and then dried at 70 °C for 3 h. For comparison, samples with different dosages of Co(NO_3_)_2_·6H_2_O and Fe(NO_3_)_2_·9H_2_O were prepared using the same procedures. 

### 2.4. Electrochemical Measurement

The electrochemical measurements were conducted on the electrochemical workstation (CHI 660E) with a standard three–electrode system. A graphite electrode was used as the counter electrode. The Hg/HgO and saturated calomel electrode (SCE) were used in 1 M KOH solution and 1 M PBS solution, respectively. CoFeNi–MOF/NF (1 cm × 1 cm) was used as the working electrode. All potentials reported in this work were converted to reversible hydrogen electrode (RHE) according to the following equations: $$E_{RHE}=E_{Hg/HgO}+0.098\;V+0.059\times pH$$
or
$$E_{RHE}=E_{SCE}+0.244\;V+0.059\times pH$$

Linear sweep voltammetry (LSV) tests were carried out at the scanning rate of 5 mV s^−1^, and an iR correction was carried out at 85% to ignore the influence of uncompensated resistance. 

### 2.5. Characterizations

The morphologies of samples were analyzed using a field emission scanning electron microscope (FESEM) SIGMASEM300 (ZEISS, Oberkochen, Germany). Energy Dispersive Spectroscopy (EDS) was performed using Oxford instruments Xplore equipped with FESEM for elemental mapping. Inductively Coupled Plasma Optical Emission Spectrometer (ICP–OES) was performed using 5110 ICP–OES (Agilent, Beijing, China). X–ray diffraction (XRD) were obtained using XRD–6100 (Shimazu, Kyoto, Japan) diffractometer with Cu–Kα radiation (λ = 1.5406 Å) at 40 kV and 30 mA from 5° to 80°. High–resolution transmission electron microscope (HRTEM) was performed using JEM F200 (JEOL, Tokyo, Japan), and the EDS was performed using JED–2300T. X–ray photoelectron spectroscopy (XPS) measurements were performed on a Thermo Scientific K–Alpha photoelectron spectrometer (Thermo Fisher Scientific, Waltham, MA, USA) using Al as the excitation source, and the binding energy peak of C 1s at 284.8 eV was used as a calibration peak.

## 3. Results and Discussion

Scanning electron microscopy (SEM) images revealed that the samples exhibited consistent growth of bowknot flakes on NF (as shown in Figure 1a), which facilitates a plethora of active sites that are conducive for water splitting. Analogous morphologies of LDH can be found in Appendix A, and the larger magnification of Co_0.45_Fe_0.45_Ni_0.9_–MOF/NF is shown in Appendix A. In the solvothermal process, Co^2+^, Ni^2+^, and Fe^3+^ in the solution first replaced H^+^, which uniformly dispersed on both sides of CoFeNi–LDH, and then acted as the reaction site of MOF self–assembly, transforming all the remaining metallic oxide into a homogeneous MOF phase [35]. The LDH morphologies corroborate the efficacy of the shape–preserving method in synthesizing MOFs, as evidenced by the recurring bowknot flake configuration. Morphological variations with different Co and Fe molar ratios in the MOFs are presented in Appendix A. It is worth highlighting that variations in Co and Fe contents significantly modulate the morphologies of samples. There is a perceptible transition from a dispersed sheet–like structure to a more aggregated columnar one, potentially influencing the exposure level of the active sites. Energy dispersive spectrum (EDS) elemental mapping for Co_0.45_Fe_0.45_Ni_0.9_–MOF/NF is portrayed in Figure 1b–f, and the accompanying EDS spectra in Appendix A confirm the uniform distribution of C, O, Ni, Co, and Fe elements within the sample. To further determine the composition of metallic elements in catalyst samples, ICP–OES tests were performed as shown in Appendix A. In the synthesis process, NF itself also plays a role as one of the sources of Ni due to the etching of NF during the reaction processes by the solvent. Therefore, the percentage of Ni is much higher than that of Co and Fe, about 75% [36,37]. The metal percentages of Co and Fe are 14.26% and 10.76%, respectively, and this slight difference might be due to differences in the hydrolysis rate of the two cations [38].

We subsequently employed X–ray diffraction (XRD) to probe the structural attributes of Co_0.45_Fe_0.45_Ni_0.9_–LDH powder, Co_0.45_Fe_0.45_Ni_0.9_–MOF powder, and Co_0.45_Fe_0.45_Ni_0.9_–MOF/NF. As illustrated in Figure 2a, distinct peaks at approximately 2θ of 11.6°, 23.4°, 33.2°, 34.1°, 35.1°, 38.7°, 52.4°, 56.1°, 59.1°, and 60.6° correspond to (0 0 3), (0 0 6), (1 0 1), (0 1 2), (0 0 9), (0 1 5), (1 0 10), (0 1 11), (1 1 0), and (1 1 3) planes of CoFe–LDH (Co_5.84_Fe_2.16_(OH)_16_(CO_3_)_1.08_·0.32H_2_O PDF#050–0235) and NiFe–LDH (Ni_5.64_Fe_2.36_(OH)_16_(CO_3_)_1.18_·7.52H_2_O PDF#51–0463). In the octahedrally coordinated LDH, metal cations occupy the coordination center, and the apex of the octahedron contains hydroxide ions. During the hydrolysis process, urea was used to adjust the pH value required for the growth of LDH; meanwhile, the formed carbonate ions were supplied as the intercalation. The cationic charge in the LDH layer is compensated by the presence of hydrated and carbonate anions between the stacked layers [39]. Here, Fe^3+^ and Co^2+^ can replace some of the lattices in Ni(OH)_2_ (PDF#73–1520) to form a stable LDH structure, with excess cations balanced by anion intercalation between the hydroxide layers, and eventually generate a uniform LDH compound [27,32,40]. The XRD for Co_0.45_Fe_0.45_Ni_0.9_–MOF/NF presented in Figure 2b reveals that, due to the pronounced diffraction peaks from the NF substrate, the characteristic peaks of the MOF are not obvious, but still can see a weak peak at 2θ of 8.9° (as shown in the partially enlarged image in Figure 2b). However, the analysis of the powdered samples indicates that peaks at 2θ of 8.9°, 14.9°, 15.7°, and 16.9° are all corresponding to the crystal facets of Ni–MOF (CCDC no. 985792), Co–MOF, and Fe–MOF [41,42]. Importantly, a slight leftward shift from the standard diffraction peaks was observed, signifying that Fe and Co doping induces lattice expansion in the resultant sample [43]. Appendix A shows the HRTEM and EDS results of Co_0.45_Fe_0.45_Ni_0.9_–MOF/NF. Although XRD results show that we have successfully synthesized the crystal structure, it is difficult to observe the obvious lattice fringes in HRTEM images because the lattice structure of MOF is easily decomposed under the electron beam, as shown in Appendix A [44,45]. Appendix A show the EDS results of Co_0.45_Fe_0.45_Ni_0.9_–MOF/NF, indicating the uniform distribution of Co, Fe, and Ni in the catalyst.

X–ray photoelectron spectroscopy (XPS) was employed to delve deeper into the elemental composition and associated chemical states of samples. Figure 3a verifies the existence of C, O, Ni, Fe, and Co elements in Co_0.45_Fe_0.45_Ni_0.9_–MOF/NF, and the Ni 2p peak of Co_0.45_Ni_0.9_–MOF/NF is relatively lower than that of Co_0.45_Fe_0.45_Ni_0.9_–MOF/NF and Fe_0.45_Ni_0.9_–MOF/NF. XPS peak strength is related to the relative content of elements. However, because the amount of metal salt used in the synthesis process and the element content in the final synthesized catalyst are not the same, the peak strength of the same element in different materials might be different. Moreover, the absence of Fe in Co_0.45_Ni_0.9_–MOF/NF reduces the etching of NF by Fe hydrolysis during the synthesis process, which is consistent with what has been reported in the literature [36]. The core–level binding energy (BE) of C 1s in Figure 3b aligns with the carbon–carbon double bond (C=C) at 284.8 eV, associated with the benzene rings from the H_2_BDC linker, the C–O polar covalent bond at 286.3 eV, and the carboxylate groups (O–C=O) of the organic ligand at 288.4 eV. The BE diagram of O 1s in Figure 3c displays peaks at 529.5 eV, 531.1 eV, 531.9 eV, and 533.0 eV and are attributed to Metal(M)–O, M–OH, oxygen vacancy, and O–C=O, respectively. The high–resolution spectra of Co 2p, Fe 2p, and Ni 2p split into 2p_1/2_ and 2p_3/2_ due to spin–orbit coupling. Specifically, for Co_0.45_Fe_0.45_Ni_0.9_–MOF/NF, the Ni 2p spectrum (Figure 3d) reveals peaks at 873.7 eV and 856 eV, corresponding to Ni 2p_1/2_ and Ni 2p_3/2_. Satellite peaks around these peaks indicate the presence of Ni^2+^ and Ni^3+^. For Fe 2p (Figure 3e), the binding energies at 724 eV and 712.5 eV are linked to Fe 2p_1/2_ and Fe 2p_3/2_, with an accompanying satellite peak signifying the existence of both bivalent and trivalent Fe. In the Co 2p spectrum (Figure 3f), peaks at 781.3 eV and 796.9 eV correspond to Co 2p_1/2_ and Co 2p_3/2_. Satellite peaks surrounding them confirm the presence of both Co^2+^ and Co^3+^ [18,27,32,40,46,47]. Due to the introduction of Fe species, the Co 2p_3/2_ peak of Co_0.45_Fe_0.45_Ni_0.9_–MOF/NF was positively shifted by 0.27 eV, and the Co 2p_1/2_ peak was negatively shifted by 0.47 eV compared to Co_0.45_Ni_0.9_–MOF/NF. Also, because of the introduction of Co species, the Fe 2p_3/2_ peak of Co_0.45_Fe_0.45_Ni_0.9_–MOF/NF was positively shifted by 0.35 eV and the Fe 2p_1/2_ peak was negatively shifted by 0.57 eV compared to Co_0.45_Ni_0.9_–MOF/NF. The above results indicate that the introduction of Fe and Co affected the oxidation states, which may influence the band structure of the elements, and thus would promote the surface charge transfer among elements, resulting in a positive impact on the catalytic effect [48]. Moreover, compared with Co_0.45_Ni_0.9_–MOF/NF, the Ni 2p of Co_0.45_Fe_0.45_Ni_0.9_–MOF/NF has a negative shift of 0.27 eV, and the overall analysis shows that the 2p orbitals of Fe and Co also have different degrees of negative shift, indicating that the presence of high–valence Co, Fe, and Ni increases, which also has a positive effect on the water decomposition reaction [49]. By comparing the semi–quantitative peak area, it is found that the proportion of Co^3+^/Co^2+^ in Co_0.45_Fe_0.45_Ni_0.9_–MOF/NF is about 10% higher than that in Co_0.45_Ni_0.9_–MOF/NF. Considering the sequence of ionic electronegativity (Co^2+^ (9.10) < Ni^2+^ (9.60) < Fe^3+^ (15.38)), the introduction of Fe can attract electrons of Co and Ni, thereby improving the adsorption and desorption ability of intermediate products in the catalytic process [40,50,51].

The OER performance of Co_0.45_Fe_0.45_Ni_0.9_–MOF/NF was assessed within a three–electrode system containing a 1 M KOH solution. Evaluations against samples with varying Co and Fe doping concentrations substantiated that the catalyst achieved peak activity when both Co and Fe were maintained at 0.45 mmol. For comparative insights, Fe_0.45_Ni_0.9_–MOF/NF and Co_0.45_Ni_0.9_–MOF/NF were analyzed under identical conditions. It can be seen from Figure 4a that Co_0.45_Fe_0.45_Ni_0.9_–MOF/NF exhibits superior OER efficacy compared to other catalysts. The OER overpotentials were only 244 mV at 50 mA cm^−2^ and 287 mV at 100 mA cm^−2^, respectively. However, Fe_0.45_Ni_0.9_–MOF/NF and Co_0.45_Ni_0.9_–MOF/NF require higher overpotentials of 260 mV and 330 mV at the current density of 50 mA cm^−2^ and 300 mV and 373 mV at the current density of 100 mA cm^−2^, respectively. Both NF and MOFs contain Ni elements, in order to prove whether the active sites of Ni in NF or the MOFs improves the catalytic activity, we compared the catalytic performance of bare NF and MOFs. The results showed that the overpotential of MOFs was much lower than that of NF (448 mV @ 50 mA cm^−2^, 491 mV @ 100 mA cm^−2^), which proved that adding Ni metal salt in the synthesis process could improve the catalytic effect. LSV tests with diverse Fe and Co doping levels are further detailed in Appendix A, which proves that Co_0.45_Fe_0.45_Ni_0.9_–MOF/NF possesses superior performance (Appendix A) among these catalysts. In addition, the overpotentials of alkaline OER of Co_0.45_Fe_0.45_Ni_0.9_–LDH/NF and Co_0.45_Fe_0.45_Ni_0.9_–MOF/NF were compared. As shown in Appendix A, the overpotentials of Co_0.45_Fe_0.45_Ni_0.9_–LDH/NF at 50 mA cm^−2^ and 100 mA cm^−2^ were 280 mV and 313 mV, respectively, both of which were significantly higher than that of Co_0.45_Fe_0.45_Ni_0.9_–MOF/NF, indicating that MOF has higher catalytic activity than that of LDH. The OER overpotential of Co_0.45_Fe_0.45_Ni_0.9_–MOF/NF under alkaline conditions compared to other catalysts is listed in Appendix A, where it competes favorably, even against other top–tier catalysts. 

Figure 4b reveals that the Tafel slope of Co_0.45_Fe_0.45_Ni_0.9_–MOF/NF (127 mV dec^−1^) is considerably lower than that of NF (157.4 mV dec^−1^), Fe_0.45_Ni_0.9_–MOF/NF (140 mV dec^−1^), and Co_0.45_Ni_0.9_–MOF/NF (151.5 mV dec^−1^), suggesting an optimal reaction kinetics in CoFeNi–MOF/NF. Figure 4c shows the electrochemical impedance spectroscopy (EIS) readings for Co_0.45_Fe_0.45_Ni_0.9_–MOF/NF, conducted at 1.5 V vs. RHE within the 10^5^–0.01 Hz frequency range, with insets of an impedance spectrum and an equivalent circuit diagram. It can be seen from Figure 4c that series resistance (R_s_ = 1.686 Ω) and charge transfer impedance (R_ct_ = 0.885 Ω) of Co_0.45_Fe_0.45_Ni_0.9_–MOF/NF are notably lower than those of other catalysts, suggesting enhanced electrical conductivity and rapid surface charge transfer [52].

Given that a catalyst’s electrochemically active surface area (ECSA) is directly proportional to its double–layer capacitance (C_dl_), the ECSA can be estimated by measuring the C_dl_ of a sample [53,54]. Figure 4d shows that the C_dl_ of Co_0.45_Fe_0.45_Ni_0.9_–MOF/NF (2.18 mF cm^−2^) surpasses those of Fe_0.45_Ni_0.9_–MOF/NF (2.09 mF cm^−2^), Co_0.45_Ni_0.9_–MOF/NF (1.44 mF cm^−2^), and NF (1.05 mF cm^−2^). This signifies that Co_0.45_Fe_0.45_Ni_0.9_–MOF/NF offers a larger active surface area for water electrolysis within a fixed catalyst geometric area [8,55]. SEM morphology of Co_0.45_Fe_0.45_Ni_0.9_–MOF/NF in Figure 1a corroborates the hypothesis that the bowknot flake microstructure exhibits a larger specific surface area, thus making more active sites accessible. This, coupled with the ECSA–normalized LSV curves in Appendix A, highlights the exceptional site activity compared to other catalysts. Notably, the formula applied to calculate ECSA is $ECSA=\frac{C_{s}}{C_{dl}}$, with C_s_ derived from an established value of 0.05 mF cm^−2^ [8,18].

The stability of the electrocatalyst is one of the most important characteristics for practical application. In this work, the stability of the catalyst was estimated by recording the change in current density with time at constant potential (as shown in Figure 4e). After the 10 h stability test, the current density of Co_0.45_Fe_0.45_Ni_0.9_–MOF/NF did not change significantly compared to the initial current density of 50 mA cm^−2^, indicating excellent stability in OER. In addition, it can be seen from Figure 4f that the catalytical performance of Co_0.45_Fe_0.45_Ni_0.9_–MOF/NF had a slight improvement after the stability test, which might be attributed to the morphology change in the catalyst before and after the stability test. The SEM image in Appendix A indicates an altered microstructure after the stability test. The morphology of the catalyst became sparser than that of before the test, and the needle–like and flaky structures coexist, potentially leading to enhanced exposure and activation of intrinsic active sites [18]. As shown in Appendix A, EDS analysis was performed on the sample after OER i–t test under 1 M KOH. The results showed an increase in the percentage content of both Fe and Ni, which might be due to the change in morphology that allows the electron beam to hit deeper material. The content of C decreased, the content of O increased, and the LSV performance was better after the stability test, indicating that the real electrocatalytic active component in the material might be the metal hydroxide or oxyhydroxide generated by the electrochemical reconstruction of crystalline Co_0.45_Fe_0.45_Ni_0.9_–MOF/NF [46,56,57].

The electrochemical performance of the catalysts was estimated in 1 M phosphate–buffered saline (PBS) solution to discern their OER catalytic prowess under neutral conditions. The disparities among these catalysts with varied molar ratios of Fe and Co were estimated as shown in Appendix A. It can be seen that the overall current density for all samples remains subdued due to the limited ion concentration demanded by reaction intermediates; however, the Co_0.45_Fe_0.45_Ni_0.9_–MOF/NF emerges as the standout performer in this cohort.

It can be seen from Figure 5a that only 505 mV was required for Co_0.45_Fe_0.45_Ni_0.9_–MOF/NF to drive a current density of 10 mA cm^−2^, which is much smaller than that of Fe_0.45_Ni_0.9_–MOF/NF (625 mV) and Co_0.45_Ni_0.9_–MOF/NF (565 mV). Benchmarking against other operational catalysts in Appendix A, the OER performance of Co_0.45_Fe_0.45_Ni_0.9_–MOF/NF under neutral conditions remains commendable. Interestingly, under 1 M PBS condition, as opposed to the 1 M KOH setting, the OER performance of Co_0.45_Ni_0.9_–MOF/NF outperforms that of Fe_0.45_Ni_0.9_–MOF/NF, emphasizing the enhancement of OER performance under neutral conditions when Co juxtaposes with Fe. This pronounced uplift in the OER performance of Co_0.45_Fe_0.45_Ni_0.9_–MOF/NF might be ascribed to the synergistic effect of Fe and Co with Ni, amplifying the inherent activity of the active sites [58]. 

Supportive evidence of superior OER reaction kinetics of Co_0.45_Fe_0.45_Ni_0.9_–MOF/NF under neutral conditions is further substantiated by the Tafel slope in Figure 5b and the EIS curve presented in Figure 5c. Besides its kinetic prowess, Co_0.45_Fe_0.45_Ni_0.9_–MOF/NF exhibits stellar stability during OER catalysis under neutral environments. Figure 5d,e highlight the near–identical current density of Co_0.45_Fe_0.45_Ni_0.9_–MOF/NF before and after a 10 h stability assessment. The congruence of LSV curves before and after the i–t test ratifies its electrochemical stability. However, the SEM image (as shown in Appendix A) reveals that the bowknot flake morphology was changed to aggregated rod–like morphology, which might account for the marginal dip in performance, possibly due to diminished exposure of the active sites. EDS analysis was also performed on the sample after the OER i–t test under 1 M PBS as shown in Appendix A. The proportions of Fe and Ni were almost consistent with the samples before the stability test, while the proportions of Co and O increased. It is speculated that, under neutral conditions, Co is the main active site in Co_0.45_Fe_0.45_Ni_0.9_–MOF/NF, which is consistent with the test results in Figure 5a, and then, it is slowly converted into metal oxide in the i–t test [59].

To encapsulate, Figure 5f consolidates the OER overpotentials of 1 M KOH and 1 M PBS for each comparative material, unambiguously spotlighting the unmatched superiority of Co_0.45_Fe_0.45_Ni_0.9_–MOF/NF across diverse testing environments and current densities.

Encouraged by the remarkable OER performance of MOFs under both alkaline and neutral conditions, we assessed the HER catalytic capabilities of these catalysts under both environments, aiming to determine their suitability as bifunctional catalysts. Preliminary experimental results with different Co and Fe molar ratios are shown in Appendix A. 

Under 1 M KOH environment, Co_0.45_Fe_0.45_Ni_0.9_–MOF/NF requires the lowest overpotential of 164 mV to achieve a current density of 10 mA cm^−2^ among these catalysts, as shown in Figure 6a. Similarly, the HER performances of Co_0.45_Fe_0.45_Ni_0.9_–LDH/NF and Co_0.45_Fe_0.45_Ni_0.9_–MOF/NF under 1 M KOH were compared. As shown in Appendix A, LDH needs an overpotential of 225 mV to drive the current density of 10 mA cm^−2^ in HER catalysis, which has a significant disparity with that of Co_0.45_Fe_0.45_Ni_0.9_–MOF/NF, demonstrating once again the performance advantage of MOF in electrocatalysis. Moreover, the Co_0.45_Fe_0.45_Ni_0.9_–MOF/NF with a Tafel slope of 91.6 mV dec^−1^, Fe_0.45_Ni_0.9_–MOF/NF of 123.3 mV dec^−1^, Co_0.45_Ni_0.9_–MOF/NF of 100.3 mV dec^−1^, and NF of 134 mV dec^−1^ shown in Figure 6b all conform to the Volmer–Heyrovsky rate–determining step [60,61]. Even though the HER stability of Co_0.45_Fe_0.45_Ni_0.9_–MOF/NF is not as extraordinary as that of OER, it still shows commendable stability after the 10 h i–t test without a notable current density drop (as shown in Figure 6c). Figure 6d,e show the LSV curves and Tafel plots of HER under 1 M PBS environment. It can be seen from Figure 6d,e that Co_0.45_Fe_0.45_Ni_0.9_–MOF/NF requires the lowest overpotential of 241 mV to drive a current density of 10 mA cm^−2^ and exhibits a higher reaction kinetics rate. Appendix A offer a comparative lens into the HER performance between Co_0.45_Fe_0.45_Ni_0.9_–MOF/NF and other studies under alkaline and near–neutral conditions, respectively. Undoubtedly, Co_0.45_Fe_0.45_Ni_0.9_–MOF/NF consistently outperforms its counterparts in HER under varying pH conditions, displaying unmatched kinetic advantages. 

Additionally, HER Nyquist plots across different environments in Appendix A also reiterate that Co_0.45_Fe_0.45_Ni_0.9_–MOF/NF has minimal resistance and superior charge transfer rates. Stability testing results (Figure 6f) demonstrate the capability of Co_0.45_Fe_0.45_Ni_0.9_–MOF/NF to execute HER performances under neutral environments with unwavering consistency. The SEM images of the catalysts after the HER tests under both 1 M KOH and 1 M PBS are shown in Appendix A, respectively, where the morphology of the aggregate block conceals the active site, and this might explain the marginal performance decline. Collectively, this evidence proves that Co_0.45_Fe_0.45_Ni_0.9_–MOF/NF might serve as a versatile bifunctional catalyst, excelling in both OER and HER under alkaline and neutral environments.

The accumulated experimental evidence unequivocally demonstrates the superior catalytic prowess of Co_0.45_Fe_0.45_Ni_0.9_–MOF/NF in both OER and HER processes, spanning alkaline and neutral environments. Motivated by these findings, we proceeded to assemble an overall water splitting cell, designating Co_0.45_Fe_0.45_Ni_0.9_–MOF/NF as both the anode and cathode, which was conducted via a two–electrode system. 

As shown in Figure 7a, the LSV curve resulting from the Co_0.45_Fe_0.45_Ni_0.9_–MOF/NF ║ Co_0.45_Fe_0.45_Ni_0.9_–MOF/NF configuration for overall water splitting highlights an impressive efficiency under alkaline conditions. A small working voltage of 1.59 V is sufficient to drive a current density of 10 mA cm^−2^. When compared to a slew of competing catalysts, the performance of this catalyst is prominently elite, as evident from Appendix A. Furthermore, this dual–functional electrocatalyst exhibited exceptional durability. It was found that there was no observable performance decline compared to the initial LSV after consistently operating at a current density of 10 mA cm^−2^ over a spanning time of 10 h. 

Our exploration continued by employing the catalyst as both an anode and a cathode in a 1 M PBS system, as illustrated in Figure 7b. Although the potential needed to achieve a current density of 10 mA cm^−2^ is elevated to 1.97 V compared to its alkaline counterpart, the catalyst still displayed commendable stability over a 10 h span. The LSV curves, before and after the durability tests, remained strikingly congruent, further bolstering its credentials as a versatile dual–function catalyst for neutral water electrolysis.

## 4. Conclusions

In the pursuit of a sustainable and environmentally friendly synthesis strategy of electrocatalyst for water splitting, this work successfully prepared a catalyst of Co_0.45_Fe_0.45_Ni_0.9_–MOF/NF via the shape–preserving two–step hydrothermal method. A distinctive bowknot flake–like structure was successfully obtained on NF, which increased the electrocatalytic surface area significantly, leading to a pronounced increase in the effective electrocatalytic active sites. Our comprehensive investigations highlighted the exceptional performances of this catalyst in oxygen evolution reaction (OER) and hydrogen evolution reaction (HER) under both alkaline and neutral conditions. The strategic synergy of Co and Fe within this framework not only improved the intrinsic activity at the catalytic sites but also accelerated the electron transfer. These combined attributes make a catalyst to exhibit commendable current densities, minimized overpotentials, and rapid charge transfer dynamics, setting it apart in terms of efficiency and stability. Furthermore, the Co_0.45_Fe_0.45_Ni_0.9_–MOF/NF catalyst exhibits superior properties as a bifunctional catalyst in overall water splitting, a pivotal step towards holistic renewable energy solutions. The innovative shape–preserving synthesis method introduced in this research sets a benchmark, offering valuable insights and a robust foundation for future endeavors in the realm of advanced electrocatalysts for water splitting.

## Data Availability

The data supporting the article’s findings are available from the corresponding author upon reasonable request.

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
