# Peer review of "Shape–Preserved CoFeNi–MOF/NF Exhibiting Superior Performance for Overall Water Splitting across Alkaline and Neutral Conditions"

_materials, 2024, doi:10.3390/ma17102195_

Round 1

Reviewer 1 Report

Comments and Suggestions for Authors

The authors synthesized Co0.45Fe0.45Ni0.9-MOF on nickel foams modified with CoFeNi-LDH by two-step hydrothermal synthesis and evaluated its electrocatalytic activity for HER and OER in alkaline and neutral conditions. The prepared Co0.45Fe0.45Ni0.9-MOF/NF exhibited superior catalytic performances and the possibility of application to overall water splitting.

I think that the revisions regarding the design and characterization of this catalyst before considering the recommendation to the publication.

1.       A lot of articles have reported superior electrocatalytic activities of heterometallic materials composed of Fe, Co, and Ni, suggesting that the formation of heterometallic systems is an already established technique for developing highly active catalysts. The authors should add more descriptions of the advantages and novelties of the catalysts they synthesized.

e.g.) ACS Appl. Energy Mater. 2023, 6, 9556-9567; etc… 

2.       In the introduction section, the authors have mentioned “At present, one-step hydrothermal method and annealing are ~ which hinders the prospect of commercial application.” (Lines82-85). However, the Co0.45Fe0.45Ni0.9-MOF/NF was synthesized by the two-step hydrothermal method which will consume more energy than the one-step way and possibly induce the instability of catalysts. I think that this introduction is inconsistent with the design of the catalyst preparation method used in this manuscript.

3.       I think that LDH is commonly used as an abbreviation for layered double hydroxide among researchers in the field of 2D materials, however, I recommend the authors write it without abbreviation at the first appearance in the manuscript.

4.       What is the role of the CoFeNi-LDH layer in this catalyst? Furthermore, the catalytic activities of CoFeNi-LDH/NF (without the MOF layer) and Co0.45Fe0.45Ni0.9-MOF/NF without the LDH layer should be discussed as the control experiments.

5.       According to Figure S3, the atomic ratios of Fe, Co, and Ni do not match the chemical equivalent of the metal salts used for the synthesis. Do the atomic ratios of metals in the catalysts show any batch dependency? If there were some dependencies, do they affect the catalytic activities and crystallinities?

6.       Although the name of CoFeNi-LDH suggests that a single material composed of three kinds of metal ions, powder XRD measurements (Figure 2a) reveal that CoFeNi-LDH is a mixture of three kinds of LDHs (CoFe-LDH, NiFe-LDH, and Ni(OH)2). Hence, the author should give a name describing that this LDH layer is a mixture. Furthermore, I wonder which LDHs contribute the most to the catalytic performance.

7.       I think it is hard to intense that Co0.45Fe0.45Ni0.9-MOF is a mixed metal MOF only based on the XRD and XPS results. The author should try high-resolution TEM observation and EDS mapping of the crystalline domain of Co0.45Fe0.45Ni0.9-MOF. If it is a mixed metal MOF, a periodic structure with a uniform presence of Co, Fe, and Ni will be observed.

Author Response

We thank the reviewer for considering this work superior. In this revision, we revised the manuscript according to reviewer’s good comments and constructive suggestions. According to the comments and suggestions, the manuscript has been revised point-by-point. Please see the attached file for Response to Reviewer 1.

Reviewer 2 Report

Comments and Suggestions for Authors

The manuscript title, “Shape-preserved CoFeNi-MOF/NF Exhibiting Superior Performance for Overall Water Splitting Across Alkaline and Neutral Conditions” by Gao et al did a detailed electrochemical study on Co0.45Fe0.45Ni0.9-MOF/NF electrocatalyst for hydrogen and oxygen evolution catalysis. The optimal catalyst Co0.45Fe0.45Ni0.9-MOF/NF shows enhanced catalytic performance at a high current density of 10 and 50 mA cm-2. The present manuscript is interesting and the author’s presentation and results & discussion part are reasonable. However, the authors must address the following major issues with their manuscript before publication in “Materials” journal.

1.      The authors should include the ICP-OES results to show the individual metal percentage in the catalyst.

2.      The authors should explain the use of Ni in optimal electrocatalysts, which could clearly reveal the catalytic performance when both Ni and CoFeNi-LDH/NF are present in the nickel foam.

3.      Why is the Ni 2p peak of Co0.45Ni0.9-MOF/NF very low in Figure 3a?

4.      Why is the Tafel value calculated in Figure 4b relatively low? Please also provide recent literature.

5.      In Figure S6 and S7, the authors can include the EDS spectrum to show the metal retention percentage after OER testing.

6.      To provide better information, it is necessary to consider some of the important recent studies, such as 10.1039/D3TA06745A, at suitable place in the revised Introduction and Discussion section.

Comments on the Quality of English Language

 Moderate editing of English language required

Author Response

We thank the referee for considering this work interesting. In this revision, we revised the manuscript according to referee’s good comments and constructive suggestions. According to the comments and suggestions, the manuscript has been revised point-by-point. Please see the attached file for Response to Reviewer 2. 

Round 2

Reviewer 1 Report

Comments and Suggestions for Authors

Regarding Reply 1, the authors add an explanation about the advantages of heterometallic catalysts as the revision according to Comment 1. Of course, this added description is important, however, they should also add a description of the advantages of the Co0.45Fe0.45Ni0.9-MOF/NF catalyst they developed in order to emphasize the importance of their research.

Please also check the following.

Line 96: by the sample one-step hydrothermal method -> by the simple one-step hydrothermal method?

Line 107: An error message has been remained. “Error! Reference source not found.”

The other points I mentioned in the previous review have been revised and replied to based on the comments.

Author Response

We thank the editors and reviewers for their recognition of our revision work. In response to the second round of comments and suggestions, the manuscript has been revised based on the first round of revisions. We greatly appreciate the referee’s very kind reminder. We have revised the manuscript point-by-point. Please see the attached file for the detailed response. 

Reviewer 2 Report

Comments and Suggestions for Authors

My concerns are fully addressed in the revised manuscript.

Comments on the Quality of English Language

 Minor editing of English language required

Author Response

We thank the editors and reviewers for their recognition of our revision work. In response to the second round of comments and suggestions, the manuscript has been revised based on the first round of revisions.

We greatly appreciate the referee’s very kind suggestion. In the revised version, the language of the whole manuscript has been improved.